# A new stochastic optimization technique for combating data poisoning attacks

## Abstract

In this paper, we present new techniques for building and analyzing robust stochastic optimization algorithms. To solve the given $d$-dimensional optimization problem, our technique generates a sequence of random $k$-dimensional subproblems, where $k < d$, and solves them instead. Unlike traditional optimization analysis which exploits structural assumptions like convexity, Lipschitzness or Polyak-Lojasiewicz criterion of the loss function to obtain convergence rates, our analysis only uses the geometrical structure of the randomness used in the algorithm. This offers a wider applicability than traditional methods, and indeed it applies to all smooth loss functions. Moreover, our analysis identifies an important parameter of the minimizers of the loss function, which we call the *gap parameter*. This parameter dictates the convergence rates of our algorithm. We experimentally study the algorithm on linear regression, logistic regression, SVMs, and neural networks. Using these experiments, we argue that the gap parameter of a minimizer also controls its robustness to the presence of noise in the training data (popularly referred to as *data poisoning*). A modified algorithm which can control the effect of noise on its output is presented as well. Finally, we discuss how the choice of $k$ affects the convergence and robustness of our algorithm.

## 1 Introduction

We study methods for stochastic optimization in the setting where a subset of training data might be corrupted by an adversary. Stochastic methods in optimization have become an important workhorse in the practice of modern Machine Learning. These methods usually work on data collected from a variety of different sources (such as scraping the Internet). Naturally, an adversary can inject malicious data in this training set and it might be very challenging to detect and remove this corrupted subset. In such a scenario, it is desirable to have algorithms which are immune to injections of small amount of arbitrary corruptions. In this paper, we propose a novel method for stochastic optimization which has the potential of addressing this problem for a wide class of loss functions.

**Difficulty of the problem.** The problem of learning good models under worst case corruptions in training data is NP-hard even for simple problems like binary classification with half-spaces (Guruswami & Raghavendra, 2006). The popular method of dealing with these difficulties is to make distributional assumptions over the data or the noise added by the adversary (Khardon & Wachman, 2007). These distributional assumptions heavily dictate the design of appropriate algorithms in these settings. However, one doesn't always know how the distributions will look like in practice and, moreover, in the case of noise a determined attacker might tailor it to the specific training data at hand and hence invalidate any distributional assumptions made. This makes the problem of building optimization problems which are robust to worst-case noise in the training data seemingly intractable.

In this work, we propose a tractable way out of this difficulty. Instead of expecting our algorithm to behave perfectly on *all* input instances and be able to handle *all* worst-case noise (which makes the problem NP-hard), we build an algorithm that performs well on *most* instances and handles worst-case noise on these instances without putting any distributional assumptions on either the training data or the noise. By working well on *most* input instances we expect to capture all the instances that one could reasonably expect to see in practice, while leaving out the small fraction of instances which are often responsible for the computational

hardness of a given problem (for example, the ones to which a reduction from an NP-hard problem like satisfiability might map to).

The idea is to develop an algorithm whose output does not change drastically under small perturbations in the training data. This algorithm is not designed to necessarily perform well on a small set of instances, even when some other algorithm might be able to solve these instances well. The hope is that this small set of instances is one that will not be seen in practice. Since characterizing the complexity landscape of various instances is a highly intricate and mathematically extremely challenging subject (Arora & Barak, 2009), the above discussion is meant to only give an intuitive understanding of our approach. As we shall see later in Section 6, we achieve our objective by identifying a crucial property (called the *gap parameter*) of the solutions of a given optimization problem. Our algorithm will only find solutions that have a large enough gap parameter, giving them good robustness properties under perturbations in the input data.

**Our approach.** Instead of working in the original dimension of the given optimization problem, our algorithm proceeds by solving the given problem in a sequence of random hyperplanes of a smaller dimension. These hyperplanes are defined so that each successive one contains the solution obtained from the previous one. The algorithm stops either when the improvements in the successive hyperplanes become small enough or after a chosen number of iterations, and outputs the solution obtained in the last hyperplane. See Algorithms 1 and 2.

**Significance of our approach.** Our approach to stochastic optimization algorithms has interesting connections with the theory of expander graphs. The role of the gap parameter in our analysis is akin to that of the spectral gap of the Laplacian of a graph. Under very mild assumptions on this parameter (the logarithm of this parameter has to be polynomially bounded in the dimension of the problem) we obtain a polynomially bounded runtime for our algorithm (see Theorem 3).

The second eigenvalue is an important spectral quantity for graphs (Kowalski, 2019) and our analysis shows that a similar quantity for functions dictates how fast certain stochastic optimization algorithms can converge. This is distinct from all previous analyses which rely on structural properties of the functions like convexity, Lipschitzness, or the Polyak-Lojasiewicz criterion to bound the rate of convergence. As far as we know, such a parallel between the very well-studied theory of random walks on expander graphs and stochastic optimization algorithms is entirely novel to our work.

The highlight of our analysis is Lemma 4, which is a statement about moving non-trivially away from the maximum of a function by random sampling. For a function whose domain is a Lie group which satisfies Kazhdan's Property (T), it gives a lower bound on the size of the set where the function takes a value non-trivially away from the maximum. This is a very general lemma (see Section 5) and we believe that it will find applicability in many future analyses.

**Organization of this paper.** In Section 2, we discuss existing approaches for stochastic optimization and dealing with perturbations in training data, popularly referred to as data poisoning attacks. In Section 3, we set up the notation and introduce all the concepts needed for the rest of the paper. In Section 4, we describe and discuss our stochastic optimization algorithm and give our convergence result. In Section 5, we discuss the main theoretical technique of our analysis. In Section 6, we discuss the robustness of our algorithm and demonstrate the efficacy of our approach with experiments in Section 7. Finally, in Section 8 we discuss the importance of our work and highlight the main takeaways from the paper.

## 2 Related Work

In this section, we compare our approach to stochastic optimization with existing approaches as well as discuss the literature on data poisoning attacks in Machine Learning.

**Comparison to existing stochastic techniques.** Most of the existing literature focuses on either stochastic gradient descent or it's popular variants like Adam (Kingma & Ba, 2014) and AdaGrad (Duchi et al., 2011). In stochastic gradient descent one picks a random subset of the data, computes the loss on this subset and uses the gradient of this loss to update the parameters of the model. Convergence for this scheme can be shown under assumption like strong convexity (Moulines & Bach, 2011), the Polyak-Lojasiewicz condition (Gower et al., 2021), and convergence to stationary points for non-convex functions which satisfy an ex-

pected smoothness assumption (Khaled & Richtárik, 2023). These convergence results rely crucially on the respective structural properties mentioned for the loss functions, while the randomness of picking a subset of the data usually worsens the convergence rates as compared to their deterministic counterparts (which work with the full training data in all iterations).

Our approach is fundamentally different from these approaches. Instead of subsets of the training data being the source of randomness, in our approach the randomness comes from the selection of random subspaces in which the given optimization problem is solved. In the particular case when the optimization problem is solving for optimal parameters in a Euclidean space, our method works in subspaces of the full space of the parameters. The only existing technique that has superficial similarities to this is the dropout method in deep learning (Srivastava et al., 2014). But even there one typically considers only subsets and not subspaces of the parameters. Note that the set of all subspaces of the parameter space is a much bigger space (being a smooth manifold) than the set of all of their subsets (which is a discrete set). In addition, dropout is a specialized technique that is only used in the context of deep learning.

Analytically, our analysis is dependent on the crucial fact that the space our randomness is drawn from forms a smooth manifold that it is a quotient of a compact Lie group, and in particular therefore satisfies Kazhdan's Property (T). The only assumption we need from the loss function is that it should be smooth. We do not need any other assumptions like convexity or Lipschitzness.

**Data poisoning in Machine Learning.** Many methods exist in the literature for dealing with data poisoning; see Tian et al. (2022) and Cinà et al. (2023) for excellent surveys. While there are a lot of methods which try to deal with data poisoning for specific models like linear regression, logistic regression, or neural networks, few methods exist which have general applicability. Data sanitization and some form of bagging and majority voting seem to be among these few general techniques. Data sanitization can be difficult as adversaries assemble more and more sophisticated forms of noise to make noisy data look indistinguishable from real data. Bagging and voting can decrease the amount of data available for training for a single model and can have unwanted accuracy trade-offs. Robust training, which augments the training data with poisoned instances to specifically train the model to handle such data, is another popular method to combat such attacks. All of these techniques deal with the preprocessing of the training data, and not with the actual learning process.

Techniques like Prasad et al. (2020) and Charikar et al. (2017), which deal with the learning process, have been developed in the robust statistics literature to mitigate the influence of noise in the training data. But these techniques tend to be intractable without making restrictive distributional or modeling assumptions.

Our technique, which is primarily a new optimization algorithm, can be used either as an alternative to these existing techniques or in conjunction with them to provide enhanced protection against data poisoning attacks.

## 3  Preliminaries

**Notation.** We use $G$ to represent a Lie group and $H$ to represent a subgroup of it. Moreover, we use $G/H$ to represent the quotient of $G$ w.r.t. $H$. For a treatment of Lie groups see Bump (2013). We use $O(d)$ to represent the compact orthogonal group acting on $\mathbb{R}^d$. The product group $O(k) \times O(d-k)$ can naturally be identified as a subgroup of $O(d)$. The quotient $O(d)/(O(k) \times O(d-k))$ has a natural interpretation as the set of all $k$-dimensional subspaces of $\mathbb{R}^d$. This is a well studied geometric object, popularly known as the Grassmannian (see Bendokat et al. (2024)). We denote it by $G_{k,d}$. We use the term $k$-plane to refer to a $k$-dimensional affine subspace, i.e. a $k$-dimensional hyperplane of $\mathbb{R}^d$, in the rest of the paper.

For us, $\ell : \mathbb{R}^d \to \mathbb{R}$ will be the smooth loss function we want to optimize. Here, smoothness means that $\ell$ is infinitely differentiable. We use $\eta$ with various subscripts to represent subspaces or $k$-planes of appropriate dimensions (which will be clear from the context).

**Measures on Lie Groups.** Our Lie groups, like all locally compact Lie groups, have a left-invariant Haar measure which is unique up to scaling (Nachbin, 1976). This covers a wide range of Lie groups used in applications (Gallier & Quaintance, 2020). For results regarding existence of invariant measures on compact Lie groups, their quotients (like the Grassmannians) and validity of Fubini style decompositions look at Chapter 1 of Sepanski (2006).

For a measurable subset $A$ of a given measure space we use $|A|$ to denote the measure of this set under the implied measure.

**Kazhdan's Property (T).** For a definition of this property see Section 3.1 of Rogawski & Lubotzky (1994). It is primarily defined for non-compact Lie groups. Indeed for compact Lie groups, such as we are considering in this paper, the property is trivially satisfied. We bring it up here because of its impact on expander graphs and their random walks which forms an important motivation for our work. Also, because we formulate our core lemma, Lemma 4, on non-compact Lie groups. We will only be working with the following consequence of the property in our proofs:

**Lemma 1** *[Remark 1.1.4 in Bekka et al. (2008)] Let $G$ be a locally compact Lie group that satisfies Kazhdan's property (T). Then there exists a $c > 0$ such that for all functions $f : G \to \mathbb{R}$, square integrable w.r.t. a left-invariant Haar measure and which satisfy $\int_G f = 0$, there exists a $\gamma \in G$ satisfying*

$$\|f - \gamma \cdot f\|^2 \geq c\|f\|^2$$

*where the action of $\gamma$ on $f$ is defined by $(\gamma \cdot f)(x) = f(\gamma^{-1} \cdot x)$.*

**Remark 2** *The constant $c > 0$ in Lemma 1 is only dependent on the group and is popularly referred to as the Kazhdan constant of the group. We can chose $c = 2$ for compact Lie groups. The proof of Lemma 4 in Appendix A.2 includes a proof of this fact.*

**Noise model.** We will study the robustness properties of our techniques in Section 6. We consider noise only in the training data matrix $A$. Noise may be introduced by perturbing a certain fraction of the rows of $A$ with a noise matrix $\Delta$ or by augmenting $A$ with a small number of well crafted data points. Generally, both these settings can be mathematically modelled as adding noise $\Delta$ to $A$. This setting is popularly referred to as *data poisoning*. We study the behavior of our approach as the fraction of rows that $\Delta$ corrupts increases. Note that we do not make any distributional assumptions on $\Delta$, instead we work with the worst case $\Delta$ by evaluating our technique against existing data poisoning attacks in the literature, which generate $\Delta$ with full knowledge of $A$.

## 4 Our Results

In this section, we describe our random walk, which is a stochastic optimization technique applicable to any smooth loss function. Moreover, we provide a convergence result that works in this very general setting.

### 4.1 Random Walk

The aim of any optimization algorithm is to find some critical point of $\ell$, usually one of the global minima, i.e., find an $x^* \in \mathbb{R}^d$ such that

$$x^* \in \arg\min_{x \in \mathbb{R}^d} \ell(x)$$

Assume that we are given a black-box access for solving the same problem but in a smaller-dimensional space, specifically a $k$-plane $\eta \subset \mathbb{R}^d$, i.e., we can find an $x_\eta^*$ such that

$$x_\eta^* \in \arg\min_{x \in \eta} \ell(x)$$

Our random walk is motivated by asking the question: Can we use this black box repeatedly for a sequence of $k$-planes $\eta_1, \eta_2, \ldots$ to find an $x^*$? This suggests a natural random walk as follows: start with some $x_1 \in \mathbb{R}^d$ and sample a random $k$-plane $\eta_1$ containing $x_1$; find an $x_2$ such that $x_2 \in \arg\min_{x \in \eta_1} \ell(x)$; in $i$-th step find a random $k$-plane $\eta_i$ containing $x_i$ and solve for $\arg\min_{x \in \eta_i} \ell(x)$; stop the algorithm after $N$ steps. We state this more formally in Algorithm 1.

This is a very natural random walk from computational complexity theory perspective. It leverages the ability to solve several smaller-dimensional random problems when solving a bigger-dimensional problem. This approach has been used to study other problems in the literature (see Section 10.1.2 in Arora & Barak (2009)), and has provided interesting insights in to the structure of these problems. This approach is called

---

**Algorithm 1** Our Random Walk

---

**Require:** $\ell : \mathbb{R}^d \to \mathbb{R}, x_0 \in \mathbb{R}^d, d > k > 1, T \geq 0, N \geq 0$
 1: **for** $i = 1, \ldots, N$ **do**
 2:     Sample $\eta_1, \ldots, \eta_T$ uniformly from $G_{k,d}$
 3:     $y_j \leftarrow \arg\min_{y \in x_{i-1}+\eta_j} \ell(y)$ for $j \in [T]$
 4:     $x_i \leftarrow \arg\min_{y \in \{y_1,\ldots,y_T\}} \ell(y)$
 5: **end for**
 6: **return** $x_N$

---

random self-reducibility. To the best of our knowledge, our work is the first to study this approach for optimization problems in the Euclidean space.

One of the surprising observations from the practice of modern ML is the ease of solving many seemingly intractable non-convex optimization problems. This justifies the use of a black-box to solve a problem of a smaller dimension in Algorithm 1. Since our random walk is designed to serve the dual purpose of optimization as well as learning a model robust to data poisoning, one would not be advised to use the same black-box to solve the original problem directly. The black box can be implemented with any of the existing techniques. We recommend using a technique best suited to the specific machine learning model that is being learned. We now study the convergence properties of Algorithm 1.

### 4.2 Convergence Analysis

For the convergence analysis, we need to define a few auxiliary functions. We define $L : \mathbb{R}^d \times G_{k,d} \to \mathbb{R}, M : \mathbb{R}^d \to \mathbb{R}, m : \mathbb{R}^d \to \mathbb{R}, \Theta : \mathbb{R}^d \to \mathbb{R}$ and $\theta : \mathbb{R} \to \mathbb{R}$ as follows:

$$L(x, \eta) \coloneqq \min_{y \in x+\eta} \ell(y),$$

$$M(x) \coloneqq \max_{\eta \in G_{k,d}} L(x, \eta), \quad m(x) \coloneqq \min_{\eta \in G_{k,d}} L(x, \eta),$$

$$\Theta(x) \coloneqq \frac{\|L(x, \cdot)\|_2^2}{2|M(x) - m(x)|^2}, \quad \theta(\alpha) \coloneqq \min_{x \in \{x : \ell(x) = \alpha\}} \Theta(x)$$

We call $\theta$ the **gap function** of $\ell$ and $\theta(l(x))$ the **gap parameter** of the minimizer $x$. The gap function of $\ell$ plays a crucial role in our analysis and have very close connections with the spectral gap of a Laplacian on a graph. We discuss this connection in more detail in the next section. Our main convergence proof is as follows:

**Theorem 3** *Let $\ell : \mathbb{R}^d \to \mathbb{R}$ be a smooth loss function such that $\theta(\ell) \geq 1 - \delta$ for some $\delta > 0$. Let $\alpha = \min_x \ell(x)$. For all $\epsilon_0$ and $\gamma$ in $(0, 1)$, with $N = \frac{\log 1/\epsilon_0}{\log 2/\delta}$ and $T = \frac{\log N + \log 2/3\gamma}{\log 1/\delta}$ and with probability at least $1 - \gamma$, Algorithm 1 finds an $x \in \mathbb{R}^d$ such that*

$$\ell(x) - \alpha \leq \epsilon_0(\ell(x_0) - \alpha).$$

We defer the proof of Theorem 3 to Appendix A.5 and discuss the main theoretical ideas behind it in Section 5. For now, there are several interesting points to note about Theorem 3:

1. It only uses a smoothness assumption on the loss function $\ell$. We believe that this assumption can be relaxed to a continuity assumption with a little bit more work. But for ease of exposition, we avoid it. In particular, note that we do not assume any bound on the Lipschitz constant of $\ell$, which is quite unusual for convergence analysis in the optimization literature.

2. The dependence on all parameters is logarithmic. In contrast, the dependence on the relevant parameters (like Lipschitz or Polyak- Lojasiewicz constant) is at least linear for gradient descent and its stochastic counterparts. Moreover, the dependence on $\epsilon_0$ for stochastic procedures is also always at least linear in $1/\epsilon_0$ even under very limited setting of convex functions (Garrigos & Gower, 2024).

3. The analysis is non-local in the sense that at each iteration we directly track progress with respect to the global minimum value $\alpha$. In typical analysis in the non-convex optimization literature one uses bounds on the difference between consecutive iterates, i.e, $\ell(x_i) - \ell(x_{i-1})$.

## 5 Main theoretical technique

In this section we state and discuss Lemma 4 which forms our main theoretical technique. We state Lemma 4 more generally than is needed to prove Theorem 3. It is stated for any locally compact Lie group that satisfies Kazhdan's Property (T). We do this in order to emphasize the general nature of our result and to bring out the connection of this crucial lemma with Kazhdan's Property (T) which is a very important and extensively studied property of Lie groups (Bekka et al., 2008). A proof Lemma 4 is presented in Appendix A.1.

**Lemma 4** *Let $G$ be a locally compact Lie group that satisfies Kazhdan's Property (T) with constant c. Fix a normalized left-invariant Haar-measure on $G$. Let $f : G \to \mathbb{R}$ be a smooth function such that $\int_G f = 0$. Let $\alpha = \min_{g \in G} f(g)$, $\beta = \max_{g \in G} f(g)$ and $\epsilon = \frac{c\|f\|_2^2}{2|\beta - \alpha|^2}$. Then,*

$$\left| \{g : f(g) - \alpha \le (1 - \sqrt{\epsilon})(\beta - \alpha)\} \right| \ge \epsilon/2.$$

**Contextualizing Lemma 4.** The lemma gives a non-trivial lower bound on the probability of finding a point that is substantially away from the maximum of the function defined on a locally compact group $G$, by simply sampling a point randomly according to the fixed left-invariant Haar measure. The fundamental nature of this lemma should be compared with results like the Markov inequality or the Chebyshev inequality, which give a non-trivial lower bound on the probability of getting a value close to the mean by sampling a point according to the used probability distribution.

**Generality of Lemma 4.** Though this result is stated on a Lie group one can transfer it to other spaces which lack this structure, for example, the $n$-dimensional hypercube. This is possible because one can construct a smooth map from the $n$-dimensional hypercube to the $n$-dimensional torus, which is a compact Lie group. We state this here to demonstrate the generality of Lemma 4 but we do not provide the details because we do not use such a result in the paper. In the next section, in Lemma 6, we discuss how the result can be transferred to an appropriate quotient of a Lie group. We also note that an argument similar to the proof of Lemma 4 can be constructed for a discrete group like the boolean hypercube, further increasing the applicability of our result.

### 5.1 Using Lemma 4 to prove Theorem 3

In Algorithm 1 we sample from the Grassmannian, which is a quotient space of the compact Lie group $O(d)$. We do not sample from the group directly. In Lemma 6 we show that a statement similar to Lemma 4 holds for our quotient space, also. The proof of Lemma 6 (which is presented in Appendix A.3) uses Lemma 4 adapted to the special case of compact Lie groups (presented in Corollary 5). Kazhdan's Property (T) is a concept for Lie groups and does not have an equivalent statement for their quotients.

**Corollary 5** *Let $G$ be a compact Lie group and let $f : G \to \mathbb{R}$ be a smooth function such that $\int_G f = 0$. Let $\alpha = \min_{g \in G} f(g)$, $\beta = \max_{g \in G} f(g)$ and $\epsilon = \frac{\|f\|_2^2}{|\beta - \alpha|^2}$. Then,*

$$\left| \{g : f(g) - \alpha \le (1 - \sqrt{\epsilon})(\beta - \alpha)\} \right| \ge \epsilon/2. \tag{1}$$

**Lemma 6** *Let $G$ be a compact Lie group and $H$ a closed subgroup of $G$. Let $f : G/H \to \mathbb{R}$ be a smooth function such that $\int_{G/H} f = 0$. Let $\alpha = \min_{x \in G/H} f(x)$, $\beta = \max_{x \in G/H} f(x)$ and $\epsilon = \frac{\|f\|_2^2}{|\beta - \alpha|^2}$. Then,*

$$\left| \{x : f(x) - \alpha \le (1 - \sqrt{\epsilon})(\beta - \alpha)\} \right| \ge \epsilon/2.$$

A direct proof of Corollary 5 (which also establishes Lemma 1 for compact Lie groups) is presented in Appendix A.2 and the proof for Lemma 6 is presented in Appendix A.3.

**Discussion on the gap parameter.** One of the very important application of Kazhdan's property (T) is the first explicit construction of an expander graph in Margulis (1973). By the virtue of their spectral gap (the difference between the first and second eigenvalue of the Laplacian), expander graphs have very good mixing properties, i.e., a random walk on an expander graph quickly gets distributed evenly across the graph

(Rogawski & Lubotzky, 1994). The parameter $\epsilon$ in the Lemmas 4-6 behaves very similarly to the spectral gap of an expander graph. It dictates how fast $f$ can approach its minimum $\alpha$. In fact, it plays a similar role in the proof of Theorem 3 as the spectral gap does in the rapid mixing proofs. More specifically, the key parallel with expander graphs is that their graph adjacency matrix shrinks functions which are orthogonal to constants (e.g., Lemma 1 of Miller & Venkatesan (2006)). This is the same operating principle as in Lemmas 4-6. This is the reason why we call $\theta$ the gap function of $\ell$.

## 6 Robustness

For any smooth function $\ell$, by Theorem 3 we know that Algorithm 1 converges towards its minimum $\alpha$. However, in practice, the algorithm might converge to a point different from the global minimum (see Section 7). In this section, we discuss why this can happen and what this means for the robustness of the solution obtained from Algorithm 1 under perturbations in the training data. The noise model we use in this section was described in Section 3.

### 6.1 Ignoring a small set

Let $f$ be a function on the Grassmannian. Consider the situation where there is a set $U$ of small measure on which the function dips dramatically compared to the measure of $U$. In this case, the minimum of $f$ outside of $U$ may be substantially larger than the minimum of $f$ over its entire domain. The variance of $f$ on this restricted space might still be almost the same as its variance on its entire domain. By only considering the space outside $U$, the gap parameter increases substantially. This means that the value of $f$, at a random point on the Grassmannian, will have a higher probability of being close to the minimum outside of $U$ than the one on the entire space. Mathematically, this can be formalized as follows:

**Lemma 7** *Let $G$ be a compact Lie group with a normalized measure. Let $H$ be a closed subgroup of it. Let $G/H$, the quotient of $G$ with respect to $H$, have a normalized measure on it. Let $f : G/H \to \mathbb{R}$ be a smooth function such that $\int_{G/H} f = 0$. Let $\alpha = \min_{x \in G/H} f(x)$, $\beta = \max_{x \in G/H} f(x)$ and $\alpha' \in (\alpha, \beta)$. Set $U = \{x : f(x) < \alpha'\}$ and $\epsilon = \frac{\|f\|_2^2}{|\beta - \alpha'|^2} - 2|U|\frac{|\beta - \alpha|^2}{|\beta - \alpha'|^2}$. Assume $\epsilon > 0$. Then,*

$$\left| \{x : f(x) - \alpha' \le (1 - \sqrt{\epsilon})(\beta - \alpha')\} \right| \ge \epsilon/2.$$

One way of interpreting this is that random sampling is blind to the bad behavior of the function on small sets in its domain. Leaving out the small set, we get a better gap parameter for the minimizers of our loss function $\ell$ that lie outside this set. In the next two subsections, we will discuss the implications of this for the robustness of Algorithm 1. But first we use Lemma 7 to give a new convergence result.

Define a function $\ell_{\alpha'}$ as $\ell_{\alpha'} := \max(\ell, \alpha')$ for some $\alpha' > \alpha$. With Lemma 7 in tow, we can now study the convergence properties of Algorithm 1 towards $\alpha'$ even when the algorithm uses $\ell$ in its execution. We therefore obtain the following theorem:

**Theorem 8** *Let $\ell : \mathbb{R}^d \to \mathbb{R}$ be a smooth loss function and $\alpha = \min_x \ell(x)$. Choose $\alpha' > \alpha$ and set $\ell_{\alpha'} := \max(\ell, \alpha')$. Let $\theta_{\ell_{\alpha'}}$ be the gap function of $\ell_{\alpha'}$. Assume $\theta_{\ell_{\alpha'}} \ge 1 - \delta$ for some $\delta > 0$. Then, for all $\epsilon_0$ and $\gamma$ in $(0,1)$, with $N = \frac{\log 1/\epsilon_0}{\log 2/\delta}$ and $T = \frac{\log N + \log 2/3\gamma}{\log 1/\delta}$, with probability at least $1 - \gamma$, Algorithm 1 finds an $x \in \mathbb{R}^d$ such that*

$$\ell(x) - \alpha' \le \epsilon_0 (\ell(x_0) - \alpha').$$

Note that $\ell_{\alpha'}$, as defined, might not be a smooth function. But that does not matter since we only use it to compute $\theta_{\ell_{\alpha'}}$ theoretically. It has arbitrarily close smooth approximations that yield the same $\theta$.

### 6.2 Gap parameter as a measure of robustness

In the last section, we saw that leaving a "part" of the function out can increase the gap parameter of the minimizers of the loss function $\ell$. In general, the value of $\ell$ on it's domain can vary between the maximum and minimum value of $\ell$. When set to the maximum value, the gap parameter for the corresponding solutions will be 1 and when set to the minimum value, it will have the smallest possible value for this function. We

hypothesize that for a solution $x$ returned by Algorithm 1, its gap parameter dictates its robustness as a minimizer of $\ell$.

When an adversary introduces a perturbation $\Delta$ to the data matrix, if it is able to corrupt the solutions on most of $G_{k,d}$ then the loss function is highly unstable, and there is little hope to build any protection against perturbations. But if we look at the class of loss functions for which most of this perturbation is limited to a small subset of $G_{k,d}$, then for such functions it is natural to aim to find solutions which lie outside of these easily corruptible subsets. Since, by Lemma 7, the gap parameter directly measures the size of the set that lies close to a given target value $\alpha'$, if this set is small, it makes the solutions corresponding to this target value more susceptible to noise and hence less robust. This is why it is reasonable to use the gap parameter as a measure of robustness. With this motivation we give a modification of Algorithm 1 which can be used to optimize $\ell$ up to an $\alpha'$ with a desired gap parameter. We present this in Algorithm 2 and prove that it finds the correct $\alpha'$ in Theorem 9. Note that Algorithm 2 does not need $\alpha'$ as an input parameter.

---

**Algorithm 2** Our Robust Random Walk

---

**Require:** $\ell : \mathbb{R}^d \to \mathbb{R}, x_0 \in \mathbb{R}^d, 1 < k < d, 0 < \theta_0 < 1/2, N > 0$

1: $T \leftarrow \frac{2N}{\log 1/(1-\theta_0)}$
2: **for** $i = 1, \ldots, N$ **do**
3:      Sample $\eta_1, \ldots, \eta_T$ uniformly from $G_{k,d}$
4:      $y_j \leftarrow \arg\min_{y \in x_{i-1} + \eta_j} \ell(y)$ for $j \in [T]$
5:      $x_i \leftarrow \arg\min_{y \in \{y_1, \ldots, y_T\}} \ell(y)$
6: **end for**
7: **return** $x_N$

---

**Theorem 9** *Let $\ell : \mathbb{R}^d \to \mathbb{R}$ be a smooth loss function. Then for all $N > 0$ and $0 < \theta_0 < 1/2$, Algorithm 2, with probability at least $1 - 3/2N$, converges to an $\alpha'$ with $\theta(\ell_{\alpha'}) \geq \theta_0$, i.e., it finds an $x$ such that*

$$\ell(x) - \alpha' \leq \left(1 - \sqrt{2\theta_0}\right)^N (\ell(x_0) - \alpha').$$

We provide a proof of this theorem in Appendix A.7.

### 6.3 Dependence of robustness on $k$

Up until now, we have discussed the convergence properties of the random walk, and identified the gap parameter as an important parameter controlling both the convergence and the robustness of the solution. In this section, we discuss how the choice of $k$, the dimension of the planes in which the optimization problem is solved, affects the algorithm and in turn informs the gap parameter of the solution retrieved. This subsection is best read in conjunction with Section 7 where our experimental results are presented.

A general trend in our experiments, across a range of models, is that for smaller values of $k$ the learned models usually have very good loss values and robustness properties. As $k$ increases, the loss might improve, but at the cost of decreased robustness. For example, in experiments with neural networks, the models learned with a smaller value of $k$ do drastically better on backdoor attacks than the models learned without Algorithm 1 while achieving similar accuracy to the latter on clean test data.

As $k$ decreases, the way the optimization problem is adapted to the respective Grassmannian changes, seemingly hiding solutions which are more susceptible to noise in the small sets as discussed in the last two sections. Surprisingly, the solutions retrieved still have close to optimal loss values. We believe that this robust behavior can be attributed to the difficulty of constructing perturbations which can simultaneously affect a large portion of random projections of the data matrix. Choosing $k$ appropriately, we can control the trade-off between obtaining a solution with an optimal loss value and a solution with better robustness properties.

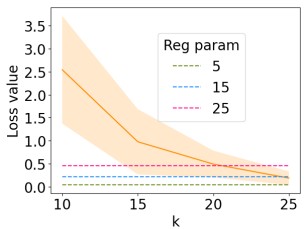 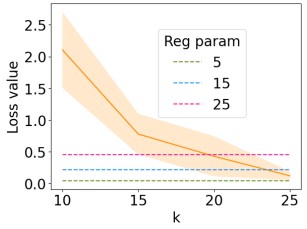 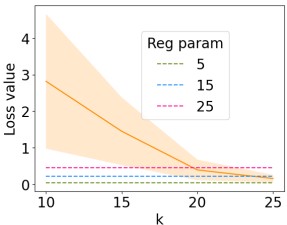

(a) Condition number $= 10^2$     (b) Condition number $= 10^4$     (c) Condition number $= 10^5$

Figure 1: Plots for Algorithm 3 run on Linear Regression. We compare the loss of the solution retrieved for different values of $k$ with the loss of the solutions retrieved by ridge regression with regularization parameters set to 5, 15 or 25. The dark lines correspond to the mean and the shaded area to one standard deviation over 10 runs of the experiment. We see that the linear regression models retrieved by Algorthm 3 have losses comparable to those of the regularized models learned with ridge regression.

# 7 Experiments

In this section, we show the versatility of our technique by testing it on a wide range of models: Linear Regression, Logistic Regression, SVMs and Neural Networks. We use both synthetic as well as popular evaluation datasets.

**Implementation details.** To simplify the implementation, we work with a modification of Algorithm 1 for our experiments. This modification is presented as Algorithm 3 in Appendix A.8. It replaces hyperplanes in Algorithm 1 with subspaces, which are hyperplanes that pass through the origin.

**Picking a random subspace.** One important step in Algorithm 3, used in all the experiments below, is that of picking a random subspace containing a given vector $x \in \mathbb{R}^d$. To do this, we consider two different techniques:

1. In the first technique, we start by constructing a basis $U$ for the space orthogonal to $x$ by taking the singular vectors corresponding to non-trivial singular values of the matrix $\mathbb{I}_d - xx^T/\|x\|^2$, where $\mathbb{I}_d$ is the $d \times d$ identity matrix. We then sample a mean 0 and variance 1 gaussian i.i.d. matrix of size $(d-1) \times (d-1)$ and construct $V \in \mathbb{R}^{(d-1)\times(k-1)}$, the matrix whose columns are the top $k-1$ left singular vectors of the randomly sampled matrix. Our desired random suspace then is the span of the column space of $UV$ combined with $x$.
2. In the second technique, we start by constructing a $d \times k$ matrix $U$ by keeping its first column as $x$ and filling the rest of it's entries with gaussian i.i.d. random variables. We then do a QR decomposition on $U$ and use the orthonormal matrix obtained from this decomposition in our algorithms. Note that the span of this orthonormal matrix will always contain $x$.

While the first method will provably generate a uniformly random subspace containing $x$, the second method has no such guarantees. But the second method is computationally much faster when $d$ is large and hence is used for all our deep learning experiments.

## 7.1 Linear Regression

For linear regression experiments, we work with synthetic data in 100 dimensions with 1000 data points. The behavior of a linear regression instance is largely determined by the condition number of its data matrix. Accordingly, we study the effect of our algorithm for data matrices with preselected condition numbers.

For a given condition number, we generate an instance whose singular values are equally spaced between a top singular value of 100 and the corresponding least singular value. We generate a regressor vector by setting the last five values to 1 and by picking other coordinate uniformly at random between 0 and 1. The idea here is that in real world data, the top singular vectors usually correspond to the signal whereas the last singular vectors correspond to the noise. We might be able to get a solution with a lower loss by fitting to the last singular vectors, but this would be overfitting to the training data. We can avoid this by using some regularization technique like ridge regression (see Section 3.4.1 in Hastie et al. (2009)). Using this setting, we want to demonstrate that for an appropriate choice of $k$, Algorithm 1 retrieves solutions which have

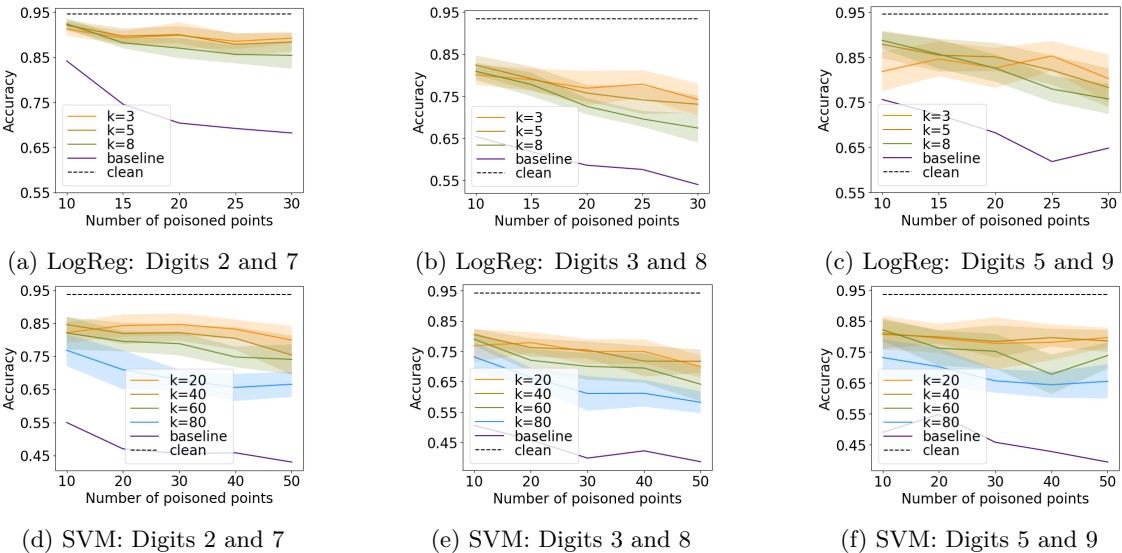

Figure 2: Plots for classifying pairs of digits from MNIST dataset using the logistic regression and SVM models trained with Algorithm 3. We poison the datasets using SecML (Melis et al., 2019) and compare the accuracy of a solution retrieved by Algorithm 3, for various values of $k$, to the solution obtained by directly learning the classifier on the poisoned dataset (this corresponds to the baseline). For reference, we also give the accuracy of the model trained on the clean data in the plots. The dark lines correspond to the mean and the shaded area to one standard deviation over 10 runs of the experiment. We see across all the plots that training with Algorithm 3 yields models with substantially better accuracy in presence of the data poisoning attacks.

loss corresponding to different choices of the regularization parameters in ridge regression. We repeated the experiments 10 times and report the mean and standard deviation in our plots. The results are presented in Figure 1. This shows that linear regression models trained with Algorithm 3, avoid fitting to the noise in the problem, and hence can be expected to have robust behavior.

## 7.2 Logistic Regression and SVMs

For binary classification experiments, we use a subset of the MNIST dataset by sampling 100 images corresponding to a pair of digits to construct our training dataset, and 500 images to construct our testing dataset. We then use SecML (Melis et al., 2019), a library for secure and explainable Machine Learning in Python, to poison the training dataset to degrade the performance of the learned classifier. The library implements the attack from Demontis et al. (2019) to generate poisoned datasets for logistic regression and the attack from Biggio et al. (2012) for SVMs. We study the effect of poisoning an increasing number of points on various choices of $k$ for Algorithm 3. As a baseline, we compare this with the accuracy obtained by training the corresponding classifiers without Algorithm 3. We also give the accuracy for training the classifiers without Algorithm 3 on a dataset with no poisoned samples. We repeated the experiments 10 times and report the mean and standard deviation in our plots. The results are presented in Figure 2. As we can see, the models obtained from the training with Algorithm 3 give much better accuracy than those trained without it. Observation also indicates that the accuracy is generally better for smaller values of $k$. We note that SVM is not a "smooth" optimization problem per se, but Algorithms 1, 2 and 3 are still well defined for it.

## 7.3 Neural Networks

In this section, we discuss the efficacy of Algorithm 3 against backdoor attacks in deep learning. The agenda of a backdoor attack is to emanate a specific response from a trained network when a test image has a special patch of pixels (the backdoor) overlapped on it. This attack can be used to misclassify images during testing. To carry out such an attack, an adversary introduces a set of images with the backdoor attached to them, and with their labels set to a desired label in the training dataset. The network then runs the risk

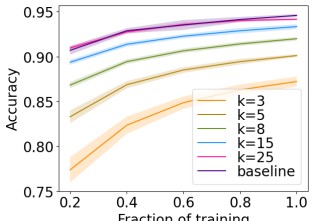 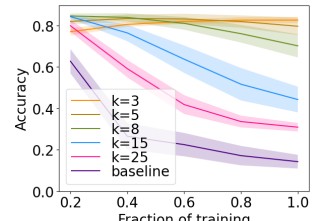 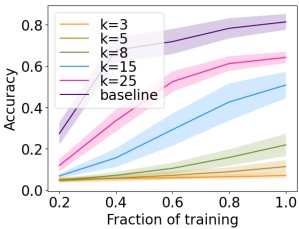

(a) Accuracy on a clean test set     (b) Accuracy on a poisoned test set    (c) Accuracy of the poisoning attack

Figure 3: Accuracy plots for MNIST against a backdoor attack presented in Turner et al. (2018). A feed-forward neural network is trained with Algorithm 3 for different values of $k$ with poisoned samples in the training data. We report three metrics: 1) accuracy on a clean test set which doesn't contain any images with the backdoor, 2) accuracy on a poisoned test set which contains images with the backdoor, 3) accuracy of the attack, i.e., images with the backdoor getting classified as intended by the adversary. We compare the results against the same model trained directly (this corresponds to the baseline). The dark lines correspond to the mean and the shaded area to one standard deviation over 5 runs of the experiment. At 0.3 fraction of the training, a modest decrease in the clean accuracy of the models trained using Algorithm 3 yields substantially better accuracy on the poisoned data set while also considerably decreasing the accuracy of the attack.

of learning an association between the backdoor and the desired label, while ignoring the true label of the image entirely.

**Attack from Turner et al. (2018).** For our experiments, we use the implementation of this attack provided in the ART toolbox (Nicolae et al., 2018). In this attack the backdoor is inserted only into images corresponding to a target label. This is done to avoid the filtering of clearly mislabeled poisoned samples by human inspection. We work with the MNIST dataset. Our model is a fully connected MLP with three hidden layers and 100 neurons in each layer. For training, we use the Adam optimizer. A baseline model is trained on the poisoned dataset for 10 epochs. For training with Algorithm 3, we set $N = 10$. To solve the problem in the subspace selected in a given iteration of Algorithm 3, we train the network for 5 epochs. The models are evaluated on a clean test set as well as a poisoned test set which consists of images corrupted with the backdoor. We repeated the experiments 5 times and report the mean and standard deviation in our plots. The results are presented in Figure 3. The baseline model corresponds to $k = 784$, which is the full dimension of the problem.

Since the parameters of an MLP are distributed across different layers and neurons, treating them as part of the same Euclidean space and working with the subspaces of this single Euclidean space is unnatural. Instead, we work with the parameters of each neuron separately by treating them as living in their own Euclidean spaces, and sampling different subspaces for each of these spaces individually when running Algorithm 3. This corresponds to working with a finite product of Grassmannians, which is still the quotient of a compact lie group (the group now will be a product of the same number of orthonormal groups). All theoretical guarantees hold in this setting, since the underlying mathematics is based on compact lie groups.

As we can see in Figure 3, the models trained using Algorithm 3 are able to achieve an accuracy on the clean test data which is close to that of the accuracy of the models trained without it. At the same time, their accuracy on the poisoned test data is substantially higher and thus the success rate of the poisoning attack substantially smaller. Specifically, around the 1/3rd training mark, the accuracy of the models trained with Algorithm 3 with $k \leq 15$ is greater than 80%, while those trained without it have an accuracy of less than 40% on the posioned test data.

**Attack from Saha et al. (2020).** For this attack too, we use the implementation provided by Nicolae et al. (2018). The intent of the attack is the same as the previous one. It is constructed in a manner so that the poisoned image is closer to the desired target image in the feature space while visually being indistinguishable from its source image. In Saha et al. (2020), the authors show that the attack is robust against many existing defense mechanisms.

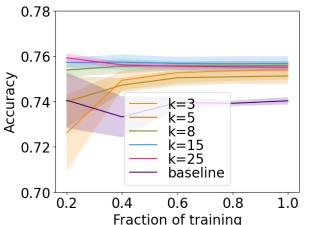 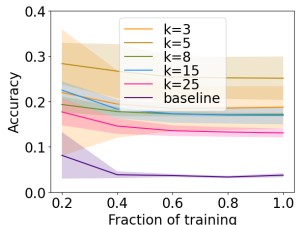 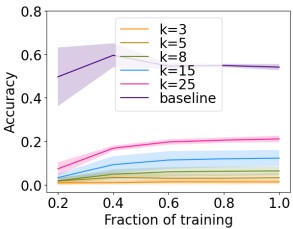

(a) Accuracy on a clean test set     (b) Accuracy on a poisoned test set     (c) Accuracy on a poisoning attack

Figure 4: Accuracy plots for CIFAR-10 against the backdoor attack presented in Saha et al. (2020). A CNN based architecture is fine-tuned with Algorithm 3 for different values of $k$ on a training set which contains poisoned data points. We report three metrics: 1) accuracy on a clean test set which doesn't contain any images with the backdoor, 2) accuracy on a poisoned test set which contains images with the backdoor, 3) accuracy of the attack, i.e., images with the backdoor getting classified as intended by the adversary. We compare the results against the same model fine-tuned directly (this corresponds to the baseline). The dark lines correspond to the mean and the shaded area to one standard deviation over 5 runs of the experiment. We see that the models trained with Algorithm 3 not only have better accuracy on the clean test set, but also have better accuracy on the poisoned test set and are able to substantially decrease the accuracy of the attack.

For our experiments, we work with the CIFAR-10 dataset and the CNNs-based architecture used by Nicolae et al. (2018) in their demonstration of the attack. We do not attempt to optimize any hyperparameters to improve the clean classification accuracy of the used model. Instead, we choose to work with the experimental setup of Nicolae et al. (2018) to demonstrate the versatility of our technique. In their setup, the poisoned dataset is used only in the fine-tuning step where all but the last fully connected layer (which has a hidden dimension of 4096) are frozen. We use Algorithm 3 on this last layer, modifying it in a way similar to what we did in the last section.

We pretrained a model for 200 epochs using SGD with learning rate 0.01, momentum 0.9 and weight decay $2 \times 10^{-4}$, reducing the learning rate by a factor of 0.1 after 100 and 150 epochs. For fine-tuning we reinitialize the last layer with gaussian i.i.d. random variables and train for another 10 epochs. For fine-tuning with Algorithm 3, apart from reinitializing the last layer, we use $N = 10$ and to solve the problem in the selected subspace of each iteration, we train for 1 epoch. We repeated the experiments 5 times and report the mean and standard deviation in our plots.

We present the results of our experiments in Figure 4 and consider three metrics: accuracy on a benign unpoisoned test set, accuracy on a poisoned test set, and the success rate of the attack on this poisoned test set. As we can see Algorithm 3 does not affect the accuracy of trained model on the benign samples, while drastically increasing its accuracy on poisoned samples and drastically decreasing the efficacy of the attack on the same samples, especially for smaller values of $k$.

## 8 Discussion

In this paper, we present a new algorithm for robust stochastic optimization. We give a general convergence theorem for this algorithm, identify an important parameter of the analysis (the gap parameter) and experimentally study the robustness properties of our algorithm. We give a modification of our algorithm which can control the robustness of its output by controlling its gap parameter, and discuss the role of the subspace dimension $k$ in our algorithm. We also present a general lemma which lower bounds the probability of the value of a function on a Lie-group, at a random point, being non-trivially away from the maximum of that function. We believe that this lemma can be adapted or applied to other settings as well.

Apart from the goal of optimization and robustness, our random walk also has the potential to provide privacy properties because of its extensive use of randomness. Studying the privacy properties of our approach is an interesting future direction. We believe that developing algorithms which can address different requirements at the same time and work for a variety of optimization problems is necessary given the recent explosion in machine learning research.

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

# A  Appendix

We present the details left out from the main body of the paper here.

## A.1  Proof of Lemma 4

**Proof.** Since $f$ is a non-constant smooth function with zero mean, we have, by Kazhdan's Property (T) that there exists a $\gamma \in G$ such that

$$\|f - \gamma \cdot f\|^2 \geq c \cdot \|f\|^2$$

where $c$ is the Kazhdan constant of $G$. Define $h : G \to [0, 1]$ as $h(g) = \frac{|f(g) - \gamma \cdot f(g)|^2}{|\beta - \alpha|^2}$. Let $U_\epsilon := \{g : h(g) > \epsilon\}$. Then, we have by Lebesgue integration,

$$\int_0^1 |U_\epsilon| d\epsilon = \int_G h(g) dg = \frac{\|f - \gamma \cdot f\|^2}{|\beta - \alpha|^2} \geq \frac{c \cdot \|f\|^2}{|\beta - \alpha|^2}. \tag{2}$$

Since $f$ is a smooth function, $|U_\epsilon|$ is a continuous non-increasing function of $\epsilon$. Moreover, $|U_0| = 1$ and $|U_1| = 0$. From this we have the following upper bound for any $\epsilon' \in [0, 1]$,

$$\int_0^1 |U_\epsilon| d\epsilon \leq \epsilon' + |U_{\epsilon'}|. \tag{3}$$

Let $\epsilon' \in [0, 1]$ be such that $|U_{\epsilon'}| = \epsilon'$. Substituting this $\epsilon'$ in (3) and using the lower bound from (2) we get $\epsilon' \geq \frac{c \cdot \|f\|^2}{2|\beta - \alpha|^2}$. For this $\epsilon'$, since $|U_{\epsilon'}| = \epsilon'$, we also have $|U_{\epsilon'}| \geq \frac{c \cdot \|f\|^2}{2|\beta - \alpha|^2}$. As $|U_{\epsilon'}|$ decreases as $\epsilon'$ increases, we can select $\epsilon' = \frac{c \cdot \|f\|^2}{2|\beta - \alpha|^2}$ and for this $\epsilon'$ we will still have $|U_{\epsilon'}| \geq \frac{c \cdot \|f\|^2}{2|\beta - \alpha|^2}$.

Now, using the definition of $h$, for every $g \in U_\epsilon$ we have

$$\frac{|f(g) - \gamma \cdot f(g)|^2}{|\beta - \alpha|^2} \geq \epsilon'.$$

On taking the denominator to the right and taking a square root on both the sides, we get either

$$f(g) - \gamma \cdot f(g) > \sqrt{\epsilon'}(\beta - \alpha) \quad \text{or} \quad \gamma \cdot f(g) - f(g) > \sqrt{\epsilon'}(\beta - \alpha).$$

Since both $f(g)$ and $\gamma \cdot f(g)$ are less than $\beta$, we can substitute $f(g)$ with it in the first equation and $\gamma \cdot f(g)$ with it in the second equation. We get

$$\beta - \gamma \cdot f(g) > \sqrt{\epsilon'}(\beta - \alpha) \quad \text{or} \quad \beta - f(g) > \sqrt{\epsilon'}(\beta - \alpha).$$

On rearranging, we get

$$\gamma \cdot f(g) < \beta - \sqrt{\epsilon'}(\beta - \alpha) \quad \text{or} \quad f(g) < \beta - \sqrt{\epsilon'}(\beta - \alpha).$$

Now subtract $\alpha$ on both sides to get

$$\gamma \cdot f(g) - \alpha < (1 - \sqrt{\epsilon'})(\beta - \alpha) \quad \text{or} \quad f(g) - \alpha < (1 - \sqrt{\epsilon'})(\beta - \alpha).$$

Since the above statements are strict inequalities, when either of them is true, there will exist a small ball around $g$ contained in $U_\epsilon$ such that the corresponding inequality will also be true for every element in this small ball. Let $U_\epsilon^1$ be the union of such balls corresponding to the set for which the first inequality is true, and let $U_\epsilon^2$ be the corresponding union for which the second inequality is true. By construction, both sets are open. Also, they are measurable as the Haar measure is a Borel measure by definition.

Now, every element of $U_\epsilon$ will belong to one of these two sets, hence $|U_\epsilon^1| + |U_\epsilon^2| \geq |U_\epsilon|$ and at least one of them will have a measure greater than or equal to $|U_\epsilon|/2$. Since the measure is left-invariant $|\gamma \cdot U_\epsilon^2| = |U_\epsilon^2|$. This gives us the conclusion. ∎

## A.2  Proof of Corollary 5

**Proof.** Consider the following integral for a non-constant zero-mean smooth function $f : G \to \mathbb{R}$,

$$
\begin{aligned}
\int_G \|f - \gamma \cdot f\|^2 d\gamma &= \int_G \int_G (f(g) - \gamma \cdot f(g))^2 dg d\gamma \\
&= \int_G \int_G \left( f(g)^2 + f(\gamma^{-1} \cdot g)^2 - 2f(g)f(\gamma \cdot g) \right) dg d\gamma \\
&\overset{(a)}{=} \int_G f(g)^2 dg + \int_G \int_G f(\gamma^{-1} \cdot g)^2 dg d\gamma - 2 \int_G f(g) \int_G f(\gamma^{-1} \cdot g) d\gamma dg \\
&\overset{(b)}{=} \int_G f(g)^2 dg + \int_G f(g)^2 dg - 2 \left( \int_G f(\gamma^{-1}) d\gamma \right) \left( \int_G f(g) dg \right) \\
&\overset{(c)}{=} 2 \int_G f(g)^2 dg \\
&= 2\|f\|^2
\end{aligned}
$$

where we change the order of integration for the last term in $(a)$, use the invariance property of the Haar measure for compact Lie groups to simplify the second and third term in $(b)$ (left invariance for the second term and right invariance for the third term) and use the fact that $f$ is mean-zero for $(c)$.

Using mean value theorem and the above calculation we see that there exists a $\gamma \in G$ such that,

$$\|f - \gamma \cdot f\|^2 \geq 2\|f\|^2. \tag{4}$$

From here on we proceed exactly as we did for the proof of Lemma 4 but with fewer details. Define $h : G \to [0,1]$ as $h(g) = \frac{|f(g) - \gamma \cdot f(g)|^2}{|\beta - \alpha|^2}$. Let $U_\epsilon := \{g : h(g) \geq \epsilon\}$. Then, we have by Lebesgue integration,

$$\int\limits_0^1 |U_\epsilon| d\epsilon = \int_G h(g) dg = \frac{\|f - \gamma \cdot f\|^2}{|\beta - \alpha|^2} \geq \frac{2\|f\|^2}{|\beta - \alpha|^2}.$$

Since, $|U_\epsilon|$ is a non-increasing function of $\epsilon$ and, $|U_0| = 1$ and $|U_1| = 0$, we have $|U_\epsilon| \geq \frac{\|f\|^2}{|\beta - \alpha|^2}$ for $\epsilon = \frac{\|f\|^2}{|\beta - \alpha|^2}$.

Now, for $g \in U_\epsilon$ we have, either

$$f(g) - \alpha \leq (1 - \sqrt{\epsilon})(\beta - \alpha) \quad \text{or,} \quad \gamma \cdot f(g) - \alpha \leq (1 - \sqrt{\epsilon})(\beta - \alpha).$$

This gives us the conclusion. ∎

## A.3  Proof of Lemma 6

We use the following lemma on the existence of an invariant measure on quotient spaces for our proof:

**Lemma 10 (Theorem 1.9 and Remark on page 93 of Helgason (2022))** *Let $G$ be a compact Lie group and $H$ a compact Lie subgroup of $G$. Then there exists a unique normalized left $G$-invariant measure $dx$ on $G/H$ such that for all $f \in C(G)$*

$$\int_G f(g)dg = \int_{G/H} \int_H f(x \cdot h)dhdx$$

*where $dg$ and $dh$ are normalized left-invariant measures on $G$ and $H$ respectively.*

**Proof of Lemma 6.** Consider any function $t : G/H \to \mathbb{R}$. Since $G = \bigcup_{h \in H} ((G/H) \cdot h)$, we can define a function $t' : G \to \mathbb{R}$ as $t'(x \cdot h) = t(x), \forall h \in H$ and $x \in G/H$. Define $f'$ using $f$ similarly. Corollary 5 gives us an estimate on the measure of the "good" subset of $G$ for $f'$. We will use $U_G$ to denote this set and $U_{G/H}$ to denote a corresponding set on $G/H$ for $f$ i.e.,

$$U_{G/H} := \left\{ x : \frac{|f(x) - \gamma \cdot f(x)|^2}{|\beta - \alpha|^2} \geq \epsilon \right\}$$

where $\epsilon$ here is the same as in Corollary 5 and, $\alpha$ and $\beta$ are the minimum and maximum values of $f$ respectively. Note that they are also the minimum and maximum values of $f'$ respectively. This follows from the definition of $f'$.

Now, let $dg, dh$ and $dx$ be normalized measures on $G, H$ and $G/H$ respectively. Using Lemma 10 we can write

$$\begin{aligned}
\int_G t'(g)dg &= \int_{G/H} \int_H t'(x \cdot h)dhdx \\
&= \int_{G/H} \int_H t(x)dhdx \\
&= \left( \int_{G/H} t(x)dx \right) \left( \int_H dh \right) \\
&= \left( \int_{G/H} t(x)dx \right).
\end{aligned}$$

Note that $f'$ is constant on the cosets $gH$ of $H$ in $G$. This means that for any $g$ in the good set of $f'$, the good set will contain the entire coset $gH$. Set $t = \mathbb{1}_{\{x \in U_{G/H}\}}$ in the above calculation, then $t' = \mathbb{1}_{\{g \in U_G\}}$. We get that measure of the good set of $f$ on $G/H$ will be the same as the measure of the good set of $f'$ on $G$. This gives us the conclusion. ∎

### A.4 Proof of Lemma 7

**Proof of Lemma 7.** We first prove an equivalent statement over the group $G$ in Lemma 11, then we can use the same machinery as we did in proving Lemma 6 from Corollary 5 to transfer the estimate from the group to it's quotient to get the full proof. The details are straightforward. ∎

**Lemma 11** *Let $G$ be a compact Lie group and let $f : G \to \mathbb{R}$ be a smooth function such that $\int_G f = 0$. Let $\alpha = \min_{g \in G} f(g)$, $\beta = \max_{g \in G} f(g)$ and $\alpha' \in (\alpha, \beta)$. Set $U = \{g : f(g) < \alpha'\}$ and $\epsilon = \frac{\|f\|_2^2}{|\beta - \alpha'|^2} - 2|U| \frac{|\beta - \alpha|^2}{|\beta - \alpha'|^2}$. Then,*

$$\left| \{g : f(g) - \alpha' \leq (1 - \sqrt{\epsilon})(\beta - \alpha')\} \right| \geq \epsilon/2$$

**Proof.** The proof proceeds in a manner similar to that of Corollary 5. We provide the extra details needed here. Define $h : G \to [0, 1]$ as $h(g) = \frac{|f(g) - \gamma \cdot f(g)|^2}{|\beta - \alpha'|^2}$ and let $V = U \cup \gamma^{-1} \cdot U$. We consider integrals over the space $G \setminus V$. To do this we use the normalized measure $dg$ on $G$ and divide it by $|V|$ so that the resulting measure is normalized over $G \setminus V$. We denote this measure by $dg_V$. We use $|\cdot|_V$ to denote the size of a set w.r.t. this measure.

Now, let $U_\epsilon := \{g : h(g) \geq \epsilon, g \in G \setminus V\}$. We use the measure $dg_V$ when measure the size of the set $U_\epsilon$. We have,

$$
\begin{aligned}
\int_0^1 |U_\epsilon|_V d\epsilon \;&\overset{(a)}{=}\; \int_{G\setminus V} h(g) dg_V \\
&= \int_{G\setminus V} \frac{|f(g) - \gamma \cdot f(g)|^2}{|\beta - \alpha'|^2} dg_V \\
&\overset{(b)}{=}\; \frac{1}{|\beta - \alpha'|^2} \left( \int_{G\setminus V} \frac{|f(g) - \gamma \cdot f(g)|^2}{|G \setminus V|} dg \right) \\
&= \frac{1}{|G \setminus V||\beta - \alpha'|^2} \left( \int_G |f(g) - \gamma \cdot f(g)|^2 dg - \int_V |f(g) - \gamma \cdot f(g)|^2 dg \right) \\
&\overset{(c)}{\geq}\; \frac{\|f - \gamma \cdot f\|^2 - |V||\beta - \alpha|^2}{|G \setminus V||\beta - \alpha'|^2} \\
&\overset{(d)}{\geq}\; \frac{2\|f\|^2 - |V||\beta - \alpha|^2}{|G \setminus V||\beta - \alpha'|^2}
\end{aligned}
$$

where we use Lebesgue integration in $(a)$, we change the measure from $dg_V$ to $dg$ in $(b)$, use the upper and lower bound on $f$ to get $(c)$ and use (4) to get $(d)$.

Since, $|U_\epsilon|_V$ is a non-increasing function of $\epsilon$ and, $|U_0|_V = 1$ and $|U_1|_V = 0$, using the same ideas as in proof of Corollary 5 we have $|U_\epsilon|_V \geq \frac{2\|f\|^2 - |V||\beta - \alpha|^2}{2|G\setminus V||\beta - \alpha'|^2}$ for $\epsilon = \frac{2\|f\|^2 - |V||\beta - \alpha|^2}{2|G\setminus V||\beta - \alpha'|^2}$. Moreover, since $|U_\epsilon|_V = \frac{|U_\epsilon|}{|G\setminus V|}$, we have $|U_\epsilon| \geq \frac{\|f\|^2}{|\beta - \alpha'|^2} - |V|\frac{|\beta - \alpha|^2}{|\beta - \alpha'|^2} \geq \frac{\|f\|^2}{|\beta - \alpha'|^2} - 2|U|\frac{|\beta - \alpha|^2}{|\beta - \alpha'|^2}$.

Now, for $g \in U_\epsilon$ we have, either

$$
f(g) - \alpha' \leq (1 - \sqrt{\epsilon})(\beta - \alpha') \quad \text{or,} \quad \gamma \cdot f(g) - \alpha' \leq (1 - \sqrt{\epsilon})(\beta - \alpha').
$$

Using the same argument as in the proof of Corollary 5 we get the conclusion. ∎

## A.5 Proof of Theorem 3

**Proof.** We use the notation set up in Section 4.2 for this proof. At step $i$ of Algorithm 1, from Lemma 6, we know that we can find an $\eta_i$ such that

$$
L(x_{i-1}, \eta_i) - m(x_{i-1}) \leq \left( 1 - \sqrt{2\Theta(x_{i-1})} \right) (M(x_{i-1}) - m(x_{i-1})) \tag{5}
$$

with probability at least $\Theta(x_{i-1})$. Since $\Theta(x_{i-1}) \geq \theta(\ell) \geq 1 - \delta$ and, as we sample $T$ points in each iteration and take the minimum over these samples, the probability that we will find one such point amplifies to $1 - \delta^T$.

The probability of this happening for all $N$ iterations of the algorithm is $(1 - \delta^T)^N$. We want this probability to be greater than $1 - \gamma$. Set $(1 - \delta^T)^N \geq 1 - \gamma$ and take the logarithm on both sides. Rearrange, and we get $N \log \frac{1}{1-\delta^T} \leq \log \frac{1}{1-\gamma}$. Now, we use the following approximations to simplify further:

$$
\forall t \in [0, 1), \quad t \leq \log \frac{1}{1 - t} \leq t + \frac{t^2}{2} \leq \frac{3t}{2}.
$$

Using these approximations it is sufficient to work with $N\delta^T \leq 3\gamma/2$. Taking logarithm on both the sides again and rearranging we get,

$$
T \geq \frac{\log N + \log 2/3\gamma}{\log 1/\delta}.
$$

This gives us a bound on the number of samples we need to draw in each iteration of Algorithm 1.

Now, we need two facts to proceed:

1. $m(x)$ is a constant function with value $\alpha$

2. $\forall i \in [1, T], \ M(x_i) \leq L(x_{i-1}, \eta_i)$.

To prove the first we proceed as follows. Recall $\alpha = \min_x \ell(x)$. Then for $k \geq 2$, $\forall x, \ m(x) = \alpha$. This is because, for any given $x$, there exists a $k$-plane that passes through $x$ and a global minimum of $\ell$.

To prove the second, notice that $x_i$ is an argmin of $\ell$ in the $k$-plane $x_{i-1} + \eta_i$. This $k$-plane will correspond to some $\eta \in G_{k,d}$ such that $x_i + \eta \cong x_{i-1} + \eta_i$. Moreover, on any other $k$-plane that contains $x_i$ the minimum value of $\ell$ will be upper bounded by $\ell(x_i)$. Hence $M(x_i) = \max_\eta L(x_i, \eta) \leq \ell(x_i) = L(x_{i-1}, \eta_i)$.

Using the above two facts we can rewrite (5) as,

$$\ell(x_i) - \alpha \leq \left(1 - \sqrt{2\Theta(x_{i-1})}\right)(\ell(x_{i-1}) - \alpha).$$

Conjugating this over all $N$ steps we get,

$$\ell(x_N) - \alpha \leq \prod_{i=1}^{N}\left(1 - \sqrt{2\Theta(x_i)}\right)(\ell(x_0) - \alpha).$$

For convenience, we loosen this equation a bit by dropping the 2 in the equation and substituting $\Theta(x)$ with $1 - \delta$ to get,

$$\ell(x_N) - \alpha \leq \left(1 - \sqrt{1 - \delta}\right)^N(\ell(x_0) - \alpha).$$

We want $\left(1 - \sqrt{1-\delta}\right)^N \leq \epsilon_0$. This gives us $N \geq \frac{\log \epsilon_0}{\log\left(1 - \sqrt{1-\delta}\right)}$. This can be further simplified as follows:

$$N \geq \frac{\log \epsilon_0}{\log(1 - \sqrt{1 - \delta})}$$
$$\overset{(a)}{\geq} \frac{\log \epsilon_0}{\log \delta/2} = \frac{\log 1/\epsilon_0}{\log 2/\delta}$$

where we use the fact that $\sqrt{1 - \delta} \leq 1 - \delta/2$ for $\delta \geq 0$ in $(a)$. This completes the proof. ∎

### A.6  Proof of Theorem 8

**Proof.** The proof here is exactly the same as the proof of Theorem 3. ∎

### A.7  Proof of Theorem 9

**Proof.** Let $\beta = \max_x \ell(x)$, then $\theta(\ell_\beta) = 1$. Now, if $\theta(\ell_\alpha) \geq \theta_0$, the theorem is trivially true. So we suppose that this is not the case. Since $\theta$ as a function of $\alpha'$ is continuous there exists an $\alpha'$ such that $\theta(\ell_\alpha') \geq \theta_0$.

Now, let $\theta_0 = 1 - \delta$. In step $i$ of Algorithm 2, with probability greater than $1 - \delta^T$ we find an $x$ such that

$$\ell(x_i) - \alpha' \leq \left(1 - \sqrt{2(1 - \delta)}\right)(\ell(x_{i+1}) - \alpha').$$

By composition, after $N$ with probability greater than $(1 - \delta^T)^N$ we have,

$$\ell(x_N) - \alpha' \leq \left(1 - \sqrt{2(1 - \delta)}\right)^N(\ell(x_0) - \alpha')$$
$$= \left(1 - \sqrt{2\theta_0}\right)^N(\ell(x_0) - \alpha').$$

Now, we need to lower bound the probability of success $(1 - \delta^T)^N$. To do so we consider the negative logarithm of this quantity,

$$-N \log \left(1 - (1 - \theta_0)^T\right) = N \log \left(\frac{1}{1 - (1 - \theta_0)^T}\right)$$

$$\overset{(a)}{\leq} \frac{3N(1 - \theta_0)^T}{2}$$

$$= \frac{3N}{2} 2^{T \log(1 - \theta_0)}$$

where $(a)$ follows from the inequality $\log \frac{1}{1-t} \leq \frac{3t}{2}$ for all $t > 0$. Setting $T = \frac{2 \log N}{\log 1/(1-\theta_0)}$, we get

$$-N \log \left(1 - (1 - \theta_0)^T\right) \leq \frac{3N}{2} 2^{-2 \log N}$$

$$\leq \frac{3N}{2} \frac{1}{N^2} = \frac{3}{2N}.$$

Hence, the probability of success is at least $2^{-3/2N} \geq 1 - 3/2N$ for all $N > 0$. This gives us the theorem. ∎

### A.8 Implementation details

---

**Algorithm 3** Random Walk for the experiments

---

**Require:** $\ell : \mathbb{R}^d \to \mathbb{R}, x_0 \in \mathbb{R}^d, 1 < k < d, N > 0$
1: **for** $i = 1, \ldots, N$ **do**
2:     $\bar{x}_{i-1} \leftarrow \frac{x_{i-1}}{\|x_{i-1}\|}$
3:     Sample $\eta$ uniformly from $G_{k-1,d-1}$
4:     $x_i \leftarrow \arg \min_{y \in \pi(\bar{x}_{i-1}, \eta)} \ell(y)$
5: **end for**
6: **return** $x_N$

---

Instead of working with Algorithms 1 and 2 as they are, we modify them a bit to make them more implementation friendly. To do this, we make two modifications:

1. First, redefine the function $L$ defined in Section 4.2 by changing its domain. To do this, define $\pi : G_{1,d} \times G_{k-1,d-1} \to G_{k,d}$ as described now. For $(x, \eta)$ in the domain of $\pi$ pick a fixed basis, represented by a matrix $U \in \mathbb{R}^{d \times (d-1)}$, for the space $x^\perp$ orthogonal to $x$ in $\mathbb{R}^d$. Note that $x^\perp$ is a $(d-1)$-dimensional space. Pick $\eta$ from $G_{k-1,d-1}$ and use a matrix $V \in \mathbb{R}^{(d-1) \times (k-1)}$ to represent it as a subspace of $\mathbb{R}^{d-1}$. Then construct a $(k-1)$-dimensional subspace of $x^\perp$ by considering the subspace spanned by $UV$. Note that this subspace will live in $\mathbb{R}^d$ and will be orthogonal to $x$. The image of $(x, \eta)$ under $\pi$ is the $k$-dimensional space spanned by $x$ and $\eta_{x^\perp}$.

   Now, define $L : G_{1,d} \times G_{k-1,d-1} \to \mathbb{R}$ as follows:

   $$L(x, \eta) := \min_{y \in \pi(x, \eta)} \ell(y)$$

2. Second, we do not sample multiple subspaces in each iteration. This decreases the computational complexity of the algorithm and is motivated by empirical observations.

We present the modification of Algorithm 1 that we use in our experiments in Algorithm 3.

The reason why we do not work with this formulation is to avoid using too many unnecessary theoretical concepts that might obscure intuition. To theoretically analyze Algorithm 3, mathematically the correct manifold to use is a degenerate flag manifold (Lakshmibai, 2009) instead of $G_{1,d} \times G_{k-1,d-1}$. The theoretical analysis still remains the same as it is mostly concerned with the use of the Grassmannian as the second space in the product manifold. However, this version of the algorithm is much easier to implement since it eliminates the affine component present in Algorithms 1 and 2.

