# OpenReview forum: "A new stochastic optimization technique for combating data poisoning attacks"
_TMLR — Rejected by TMLR_

### Review · Reviewer_CQRM · 2024-08-05

**Summary Of Contributions:**

The authors propose a method for solving a d-dimensional optimization problem by solving several k-dimensional subproblems, where k < d.  The analysis of the proposed method relies only on the randomness used in the algorithm, rather than in assumptions such as convexity or Lipschitzness.  The authors identify a parameter called "gap parameter" that governs the complexity of the proposed algorithm.  Furthermore, the authors perform experiments on linear regression, logistic regression, SVMs, and neural networks and argue that this parameter also controls the robustness of the proposed algorithm (for these models) under data poisoning attacks.  Eventually the authors discuss how the choice of k affects the convergence and the robustness of the proposed algorithm. For linear regression the authors use a synthetic dataset with varied condition number, while for the classification tasks with the other models, the MNIST and CIFAR-10 datasets are used.

**Audience:**

Yes

**Claims And Evidence:**

Yes

**Requested Changes:**

Please see the weaknesses above.

**Strengths And Weaknesses:**

**Strengths**

1. I believe that the authors have some very interesting results and a nice story to say.

2. The idea of solving a sequence of subproblems in order to solve a bigger problem is natural, interesting, and in particular allows one to find a solution to an optimization problem faster, as smaller spaces need to be explored.

3. Proposing a method for optimization that works not only in an "honest" setting but also when the training data have been manipulated by an adversary is a welcome property.

4. Arguing about the tradeoffs for different values of k is another welcome property of the paper, especially since the experiments are being conducted on important models that can be used in the real world.


**Weaknesses**

1. I am not sure if the average ML reader can appreciate the theoretical results of the paper.  For example, the authors provide guarantees that require Lie groups, the notion of the Grassmanian, and others.  Of course, formal guarantees are very much appreciated, but at the very least one needs to know what quantities map into what and why.  For example, what are Lie groups and why do we need them?  Is it natural to make such an assumption or not?  Why?  Why do these groups have to be locally compact?  Why should the reader refer to another source for "Kazhdan's Property (T)" (see, page 4, line 3)?  This is not knowledge that the vast majority of ML researchers have.  Similar for the condition number; you are using it but has never been defined.

2. Along the above lines, the preliminaries should be expanded and this way the paper can be more self-contained.  However, intuitive descriptions are required in many parts of the paper that can connect the various results with quantities that the ML research can relate to.

3. In Algorithm 1 (and Algorithm 2) I am wondering what the $\eta$'s are. In page 4 the $\eta$'s are supposed to be hyperplanes, but then what do the authors mean by adding a point $x_{i-1}$ to a hyperplane $\eta_i$?  I do not understand how this makes sense when they are trying to compute the argmin of the loss function.  Can you please explain?

4. Missing reference: in page 5, 7 lines before the end: there is an argument about the complexity of gradient descent and stochastic counterparts that is at least linear.  Please either provide a brief explanation or give a reference for the interested reader.

---

> ### Author Response · Authors · 2024-08-11
> **Highlight that the mathematical objects used in the paper are not borne of assumptions but are natural to the setting considered.**
>
> We thank the reviewer for highlighting the strengths of our paper.
> # Weaknesses:
> 1. It was our aim that an ML reader should be able to understand Alg 1, 2 and 3, and be able to appreciate the experiments, without compromising on providing theoretical details. We do appreciate that the paper relies on concepts from differential geometry which are not as familiar to the ML reader (though we note that it is also not fully unfamiliar, since ML has always had a line of papers that leveraged differential geometry). We point the reader to useful references in differential geometry that expand upon the terms and concepts we use. More specific comments are below.
> a) "what are Lie groups and why do we need them? Is it natural to make such an assumption or not? Why?"
> The use of subspaces/hyperplanes is integral to our algorithm. The set of all subspaces forms the Grassmannian. It has the property of being a quotient of a Lie group. This is how and why Lie groups enter into our analysis. We want to emphasize, however, that these are not assumptions, but are simply mathematical characteristics of the setting of the problem which we are leveraging. For any given loss function defined over the Euclidean space, one can always define the loss function in terms of the Grassmanian, as done in section 4.2. This is crucial for our analysis.
> b) "Why do these groups have to be locally compact?"
> All finite-dimensional Lie groups are locally compact, so this property is again simply a feature of the natural setting of the optimization problem. Local compactness furthermore has many important convergence properties that are tacitly assumed in iterative convergence procedures. For example, otherwise one cannot guarantee that a Cauchy sequence will converge (imagine a sequence where the distance between successive elements decreases as we go farther into the sequence, but without it ever converging).
> c) "Why should the reader refer to another source for "Kazhdan's Property (T)" (see, page 4, line 3)? This is not knowledge that the vast majority of ML researchers have."
> The reason we did not expand on it is that the formal definition is technical and is unnecessary for our results, namely: a Lie group satisfies Kazhdan’s Property (T) if its trivial representation is isolated in its unitary dual under the Fell topology. However, it’s still important to bring out the connection with Kazhdan’s Property (T), since we leverage it in our analysis as it relates to the theory of expander graphs (see the discussion at the end of Section 5). We note that Corollary 5 provides a more direct approach with a proof that is independent of this property. Our aim was to bring attention to a part of mathematics that has hitherto gone unused in ML, but might be beneficial in the future, and certainly was the motivation for our approach.
> d) "Similar for the condition number; you are using it but has never been defined."
> We will be happy to add a definition in the next version of the paper.
> 2. The use of groups and quotients, though unusual, is not entirely new to optimization. Optimization on manifolds is a well developed field, where these mathematical objects are discussed regularly. For example, see Optimization Algorithms on Matrix Manifolds by P.-A. Absil, et al. As suggested, we will be happy to add some more qualitative descriptions of these objects to Section 3, like defining a Lie group as a group which also has the structure of a smooth manifold, and a quotient of a space as the set of all equivalence classes defined by a given relation over the elements in the space. To address the other concern, we do connect our results with quantities that ML research can relate to; for example, the discussion after Theorem 3 on page 5 contrasts our results with typical convergence analysis, and the discussion after Lemma 4 connects our work with other fundamental theoretical tools like the Markov inequality (which are widely used in ML). It is difficult to directly connect the main theoretical workhorse of the paper (Lemma 4, Corollary 5, Lemma 6) with existing ML research primarily because the kind of connection between optimization and random walks that we bring out in this work has not been considered before, though at a high level other connections between these topics have been explored in the past.
> 3. As specified in the algorithm, $\eta\in G_{k,d}$, hence it is a subspace. We follow the standard mathematical terminology that the sum of a point $x$ with a hyperplane $V$ is a shifted hyperplane, namely the collection of all points $x+v$, where $v$ ranges over $V$.  In our context, this effectively moves $\eta_i$ away from the origin in the direction of $x_{i-1}$ by a distance of $\|x_{i-1}\|$, making it a hyperplane. We will be happy to add this detail to Section 3.
> 4. The reference for that is the same reference as in the end of that bullet point (Garrigos and Gower, 2024). We will be happy to rephrase the sentence to make it more explicit.

---

### Review · Reviewer_FjQF · 2024-08-22

**Summary Of Contributions:**

The authors argue that due to the function's smoothness only, a subspace with non-zero measure exists where the function is near optimal. The probability of randomly hitting this subspace depends on the gap parameter, which is central to their contribution.

**Audience:**

Yes

**Broader Impact Concerns:**

-

**Claims And Evidence:**

Yes

**Requested Changes:**

I have not exhaustively checked the mathematical details due to a lack of knowledge regarding Lie groups.

- Paragraph "Significance of our approach" assumes that the logarithm of the gap parameter has to be polynomially bounded in the dimension of the problem". Can you provide a class of functions that meet this assumption along with some examples and define the bound explicitly?

- The assumption also refers to Theorem 3, yet neither the statement nor the proof explicitly addresses the dimension of the problem. The explicit dependence should be provided and compared with the existing lower bounds (non-convex Lipschitz function, bounded domain). Could the gap parameter be so close to zero that a naive random search guarantee faster convergence in the case of a Lipschitz function on an $ l_\infty$ ball? In other words, T can be arbitrarily large without a non-zero lower bound on the gap parameter. At the very least, kindly consider providing some trivial examples.

- The notation needs improvement, such as clarifying what is denoted by $L(x, ·)$ in the definition of $\Theta(x)$ and defining $\|\|f\|\|$.

**Strengths And Weaknesses:**

### Strengths
- Novel approach
- Function to minimized is only assumed to be smooth

### Weaknesses
- Special structure of the space is assumed

---

> ### Author Response · Authors · 2024-09-19
> **Addressing the concerns highlighted by the reviewer**
>
> We thank the reviewer for all the comments and address the concerns as follows.
> # Weaknesses:
> 1. We are confused about what special structure the reviewer is referring to. Our approach works for every smooth loss function defined over the Euclidean space. Euclidean space has subspaces. The set of all subspaces of a particular dimension forms a manifold called the Grassmannian, which is a quotient of a compact Lie group. These are, for the most part, all the mathematical objects considered in our manuscript. Note that there are no special assumptions made anywhere here.
> # Requested Changes:
> 1. This parameter $\epsilon$ in Lemma 6 will be bounded by the Lipschitz constant of the function $f$ defined over the quotient space. On the other hand, the gap function of a function $\ell$ is defined using the function $L$. The definition of the function $L$ uses a minimum of $\ell$ (the original loss function) in subspaces. This definition is non-local in some sense. One cannot use properties like Lipschitzness or various characteristics of the Hessian to necessarily transfer Lipschitzness of $\ell$ to $L$. Hence, the same properties cannot be easily transferred to the gap function of $\ell$ either. However, this is not needed. As we discussed in Section 6.2 one can always ignore a part of the space and obtain a gap parameter on the rest of the space which is big enough. In this case the random walk will converge to a solution that lies outside the space ignored. It is these solutions which show good robustness properties, as we verify with the experiments. We will be happy to rewrite the passage to emphasize this nuance.
> 2. As we discuss above it is unclear how to get an explicit dependence on the dimension parameter, but at the same time it is not relevant to the analysis at hand.
> 3. The (standard) mathematical usage of $L(x,\cdot)$ we have adopted here simply means that the first parameter of the function is fixed; it is only a function in its second parameter. We will be happy to explain this in the notation section. We note that we are using the $L_2$ norm everywhere.

---

### Review · Reviewer_q9iv · 2024-08-22

**Summary Of Contributions:**

This paper presents a novel stochastic optimization algorithm for resolving the data pointing attacks in ML problems. Specifically, the authors solve the large dimensional optimization problem through a sequence of random small dimensional subproblems. They claim such a method enables the relaxation of some typical assumptions on the loss function to obtain the convergence rates. Their analysis only leverages the geometrical structure of the randomness. They also identifies an important parameter of the minimizers of the loss function, called gap parameter, which determines the convergence rates. To validate the proposed algorithm, the authors experimentally study the algorithm on linear regression, logistic regression, SVMs, and neural networks. They have found that the gap parameter also controls the robustness against the presence of noise in the training data. Given this, they also present a modified algorithm which can control the effect of the noise on its output. A discussion on the impact of the choice of $k$ on the convergence and robustness is given.

**Audience:**

Yes

**Broader Impact Concerns:**

The authors have given some discussion on the broader impact concerns in the work.

**Claims And Evidence:**

Yes

**Requested Changes:**

Please see the above comments.

**Strengths And Weaknesses:**

Strengths:

1. This study presents a novel stochastic optimization algorithm based on random walk to decompose a large problem into small subproblems
2. It presents complete theoretical convergence analysis for regular and robust ML models
3. The authors identify a key parameter to determine the convergence rate and technically discuss the impact of the parameter on robustness
4. The authors show different models and datasets to validate the efficacy of the proposed algorithms.

Weaknesses:

1. Though the authors claim that the new algorithms do not require the extra assumptions on the objective loss, it is unclear to me what the technical difference is between the proposed algorithms and the existing ones in terms of convergence rates. Are they still similar or completely different? The authors need to give more technical details to clarify
2. The experimental results are not that promising as it is lack of some popular baselines. The authors need to show the comparison between the proposed algorithms and the popular existing ones.
3. In Theorem 3, $\epsilon_0$ is a constant between 0 and 1. If $\epsilon_0$ is quite close to 1, then the optimization wouldn't proceed much, which would hurt the solution. Is there any guarantee or analysis that the authors can give to push the optimization forward and make sure the solution is decent?
4. In Theorem 9, the convergence rate seems linear. While without some further assumptions on the loss, such as strong convexity, it would be difficult to obtain such a result. I am a bit confused how the robust random walk has the same ability to enable linear convergence rate as the assumption of strong convexity on the loss function. The authors need to add more technical details to discuss this.

---

> ### Author Response · Authors · 2024-09-19
> **Refer the reviewer to appropriate parts in the paper to address the weaknesses highlighted.**
>
> We thank the reviewer for highlighting the strengths of the paper and we address the weaknesses below.
> # Weaknesses:
> 1. We refer the reviewer to the discussion after Theorem 3 in Section 4.2 where we explicitly compare our convergence rates with those of other popular techniques.
> 2. We do not present our algorithms as competing with other techniques since we do not know any other technique which deals with data poisoning attack at the level of the optimization algorithm and which works in the general setting that our technique does. Please see the discussion under “Data poisoning in Machine Learning” in Section 2.
> 3. $\epsilon_0$ is a parameter that can be chosen by the user. If the user wants to find a solution where the loss value is very close to the absolute minimum then we would recommend setting it to a small value.
> 4. Please see the discussion after Theorem 3 for a direct comparison with analysis of more popular optimization techniques. We also note that we provide full proofs for all our technical claims including those of linear convergence rates, and refer the reviewer to the appendix to see the technical details. As we note in various remarks in the paper, our analysis doesn’t proceed via the usual techniques used in the optimization literature (which work with quantities like gradient, Hessian, etc).

---

### Review · Reviewer_foEf · 2024-08-24

**Summary Of Contributions:**

The paper studies solving large-scale optimization problems with an optimization variable in $\mathbb{R}^d$ by iteratively solving the problem over random subsets in a smaller dimension $k$ (Algorithm 1). The paper has results on the convergence analysis of the proposed algorithm using an assumption on the gap function defined in section 4.2. Theorem 3 shows linear convergence to the optimal value using an $N$ hyperparameter on the order of $\log(1/\epsilon_0)$. The gap parameter is discussed in the paper and has been related to the robustness against data poisoning to the training data. The paper has numerical results on basic image classification datasets MNIST and CIFAR-10, to show the proposed approach can offer higher robustness against data poisoning.

**Audience:**

Yes

**Broader Impact Concerns:**

I do not see any major ethical concerns that remain undiscussed in the text.

**Claims And Evidence:**

No

**Requested Changes:**

Based on what I discussed on weaknesses, the following changes can improve the paper:

1. The related work discussion should discuss randomized black coordinate descent methods, as they follow the same idea as the authors' proposed method. Also, the numerical experiments should use the coordinate descent algorithm as a baseline to ensure the algorithm provides numerical improvement over standard block coordinate descent methods where the subspace is selected by choosing blocks of features.

2. The theoretical results on convergence rate should be more clearly connected to the role of subspace dimension $k$.

3. Sections 6.2 and 6.3 need revision and the writing should be more precise. The numerical and theoretical claims on the connection between gap parameter and robustness sound vague in these sections.

4. The numerical results on linear regression and SVM, representing convex optimization problems, should be better clarified, as the optimization task will have a unique locally optimal solution which is supposed to be found by a satisfactory optimization algorithm.

5. For the neural net experiments, the authors should provide the details of selecting the random subspaces.

**Strengths And Weaknesses:**

**Strengths**

1. While the paper's optimization approach is not very novel and can be thought of a randomized coordinate descent method, I find the authors' idea of connecting the gap parameter to the robustness of the learning algorithm interesting. I believe the connection can be explored more deeply and can be useful to understand the robustness properties of stochastic optimization methods.

2. The paper is generally well-written and clearly presented. It is rather easy to read and follow the discussion.

**Weaknesses**

1. The proposed approach seems not that novel and the idea of using variable subsets in the iterations of optimizing the target function has been well explored in the optimization literature. In fact, block coordinate descent methods also follow the same strategy discussed by the authors. In a standard coordinate descent algorithm, the optimizer solves the problem across one or a few coordinates at each iteration, and this algorithm is highly used and analyzed in the context of sparse and $L_1$-regularzied optimization settings. It is strange that the literature review in this paper did not cover coordinate descent methods, as they seem strongly connected to the authors' approach.

2. Based on the introduction, it seems the paper's main goal of convergence analysis is to show the relation between the choice of subpsace dimension $k$ and the convergence properties of the method. However, the theoretical results in Theorem 3, Corollary 5, Theorem 8 do not analyze the role of $k$, and it seems they look equally applicable to different values of $k$. The theoretical results should better analyze the role of $k$. Otherwise, they will be generic statements that are not very useful for analyzing the authors' proposed approach. The discussion in section 6.3 only tries to give intuition on the role of $k$ and does not contain any theoretical results.

3. While the paper provides some non-technical argument on why the gap parameter is connected to the robsutness of the algorithm (Section 6), I think the precise connection remains vague and section 6 does not give theoretical understanding of the connection. Especially, in section 6.2, the authors only discuss and defend a hypothesis as they say "We hypothesize that for a solution $x$ returned by Algorithm 1, its gap parameter dictates its robustness as a minimizer of $\ell$." This sentence and the post discussion are imprecise and lack the rigor expected from a technical machine learning or optimization work. For example, when the authors say "if it is able to corrupt the solutions on
most of $G_{k,d}$ then the loss function is highly unstable, and there is little hope to build any protection against perturbations", what does the term "little hope" precisely mean? This and similar sentences in the section have to be revised, because a machine learning paper is supposed to state more precise claims.

4. In the experiments section, the authors discuss the computational costs of randomly sampling the subspace. I think if the selection of random subpsaces is computationally heavy, then why would a practitioner use the method instead of standard SGD or mini-batch GD to solve the linear regression or logistic regression problems? As another question, the linear and logistic regression problems are convex tasks for which the algorithms with different values of $k$ are expected to converge to the same model (the global minimizer of the convex optimization problem), then what is the motivation of using the authors' approach in this setting? I mean, where the authors say "for an appropriate choice of k, Algorithm 1 retrieves solutions which have loss corresponding to different choices of the regularization parameters in ridge regression." how would it be possible for a convex optimization task like linear regression that the authors' algorithm find different solutions for different values of hyperparameter $k$? Does that mean the algorithm finds a solution that is not the global minimum of the convex loss function?

5. For the neural network experiments, the selection of the random subspace should be more clearly explained. The sentence "Instead,
we work with the parameters of each neuron separately by treating them as living in their own Euclidean
spaces, and sampling different subspaces for each of these spaces individually when running Algorithm 3" is vague and does not include the details of selecting the random subspace.

---

> ### Author Response · Authors · 2024-09-19
> **Dissimilarity between co-ordinate descent and our method and a detailed explanation of our hypothesis.**
>
> # Strengths:
> 1. We would like to highlight that our random walk (RW) is not just a randomized coordinate descent (RCD) method. In RCD one chooses a _subset_. In contrast, we choose a random _subspace_. In $\mathbb{R}^d$ one can choose $2^d$ many co-ordinate subsets (large but finite). While the set of $k$-dim subspaces of $\mathbb{R}^d$ forms a manifold (an uncountable set). We rely on the geometry of this manifold, in contrast to the finite set of subsets in RCD.
> 2. We thank the reviewer for the compliment.
> # Weaknesses:
> 1. Please see 1 above and 4 below. We have not seen RCD being studied for robustness. A robust optimization algorithm is our main goal (see paper title). Hence, we do not see the necessity for this comparison.
> 2. We do not claim an explicit connection between $k$ and the convergence rate in the abstract. In it's last line we do claim to study the connection between $k$ and convergence. Even under convexity, RW will not retrieve same/close-by solutions for different values of $k$ or on different runs (see 4 below).
> 3. To clarify highlighted statements, consider classification with data $A$. Every model in the domain has some acc associated with it. Imagine there is a small perturbation of $A$ (=$A'$) that changes the acc of most of the models dramatically (0 flips to 100, so on). This is highly undesirable. One wouldn't work with such an optimization problem to begin with. Instead, if _any_ $A'$ results in such wild changes only for a small fraction of models while affecting others in a limited manner, then there is some hope of doing robust ML. On the other hand, we show that RW can converge to some solution that lies outside a small set, which when ignored improves the gap parameter. We capture this hypothesis in Thm 9 by assuming $\theta(\ell_{\alpha’})\geq\theta_0$. We _hypothesize_ that these two small sets coincide or are at least close and justify it by experiments. Given the convergence rate of RW, it will find solutions with good gap parameters. These solutions are experimentally robust. We acknowledge the gap here and hence we explicitly mention it as a hypothesis. We don't claim mathematical rigor about this; nevertheless, it is an important hypothesis deserving future attention. This hypothesis is enabled by function $L$ defined over $\mathbb{R}^d\times G_{k,d}$. It is unclear if $\mathbb{R}^d$ can be decomposed into a product which would both enable an opt algo and an analysis similar to ours.
> 4. We will streamline the discussion of sampling subspaces. The reviewer is correct. Even under convexity (unique min), RW doesn’t yield this min. We split our explanation into two parts (in the first we do not include the role of $k$; in the second we do): a) Solving for a unique min is not an optimal strategy in adversarial settings. The intent of RW is to find a solution which minimizes the loss sufficiently while also providing robustness. As discussed in Sec 7.2, on directly using LogReg or SVM on adversarially perturbed data, the global optima defined may have very bad acc. A way to remedy this is to use regularization. Such deterministic methods leave the possibility for an adversary to design an attack against them since they know the final solution in advance. In contrast, RW is randomized. But unlike SGD/RCD it doesn't yield the global min. Instead, every run of it (even under convexity) yields a different solution. In aiming for a predefined solution, by definition the sol is available to an adversary with unlimited computational resources (enabling it to design an appropriate noise); same problem as when trying to obtain robustness by regularization. A suitable tool to address this is an appropriate use of secret randomness. This cannot be accounted for by any adversary. RW's inherent randomness is key to its robustness. We do not know of any such experimental/theoretical analysis for SGD/RCD. Practical experience suggests they do not have these properties.
> b) Now, the robustness of our RW crucially depends on the choice of $k$ (smaller yields more robust output but with a worse loss). We justify this with extensive experiments.
> 5. Consider a NN with $L$ neurons ($d$ params each) for a total of $dL$ params. We can work with $\mathbb{R}^{dL}$ and its subspaces. This would be very cumbersome to code. Instead, one can view the params as $L$ many $\mathbb{R}^d$ spaces, and pick a subspace for each space separately, simplifying the implementation. A linear transform in a neuron (a matrix-vector multiplication) can be implemented as a different linear transform plus a 1x1 convolution. We will be happy to add more explanation in the appendix.
> # Requested Changes:
> 1. The topology over the set of subsets is not suitable/adequate for the robustness results we have developed in the context of $G_{k,d}$. Apart from the many dissimilarities with RCD (see 1 above), we have not seen such algos being studied for robustness, and do not feel it is an appropriate baseline to compare against.

---

### Decision · Action_Editor_Ys4F · 2024-10-15

**Recommendation:** Reject

**Comment:**

Although the paper addresses an important problem and provides new results, it cannot be accepted in its current shape. The main reason for this is (1) the lack of technical details, (2) non-rigorous claims (e.g., see W3 provided by Reviewer foEf), and (3) the lack of in-depth comparison with the existing results in (robust) stochastic optimization.

In particular, the third issue is the most important one, in my opinion. The results depend on a new called gap parameter. However, in the paper, there is only one short paragraph about it without a clear connection to the standard quantities for (stochastic) optimization, e.g., smoothness parameter and strong convexity parameter. I believe the paper will benefit a lot from the consideration of some simple special cases, where the results can be directly compared to the existing ones. Moreover, the method studied in the paper is indeed very close to the existing methods such as Randomized Coordinate Descent, Random Direction Search [1], and Subspace Descent [2]. In addition to this, I believe the comparison with the alternating minimization procedures [3] is also required for the paper.

Nevertheless, I encourage the authors to revise their paper (in particular, address the above concerns) and re-submit either to TMLR or to other ML journal/conference.


---
References:

[1] Nesterov, Y., & Spokoiny, V. (2017). Random gradient-free minimization of convex functions. Foundations of Computational Mathematics, 17(2), 527-566.

[2] Frongillo et al, "Convergence analysis of prediction markets via randomized subspace descent", NIPS 2015

[3] Ortega, J. M., & Rheinboldt, W. C. (2000). Iterative solution of nonlinear equations in several variables. Society for Industrial and Applied Mathematics.

**Audience:**

The paper studies an interesting problem for TMLR's audience.

**Claims And Evidence:**

As some reviewers noticed in their reviews, it is hard to evaluate the contribution of the paper (and thus validate some of the claims) due to the two main reasons:
1. The paper relies on a mathematical formalism that requires a more detailed introduction for the ML paper.
2. The assumptions made in the paper are non-standard for stochastic optimization. The comparison with the existing results also has to be improved.

**Resubmission Of Major Revision:**

The authors may consider submitting a major revision at a later time.